# A protocol for a systematic review of process evaluations of interventions investigating sedentary behaviour in adults

Rekesh Corepal,[1] Jessica Faye Hall,[1] Coralie English,[2] Amanda Farrin,[3] Claire F Fitzsimons,[4] Anne Forster,[5] Rebecca Lawton,[6,7] Gillian Mead,[8] David Clarke[5]

**Correspondence to**
Dr Rekesh Corepal;
rekesh.corepal@bthft.nhs.uk

## ABSTRACT

**Introduction** Sedentary behaviour is defined as any waking behaviour characterised by low energy expenditure ≤1.5 metabolic equivalents while in a sitting, lying or reclining posture. The expanding evidence base suggests that sedentary behaviour may have a detrimental effect on health, well-being and is associated with an increased risk of all-cause mortality. We aim to review process evaluations of randomised controlled trials (RCTs) which included a measure of sedentary behaviour in adults in order to develop an understanding of intervention content, mechanisms of impact, implementation and delivery approaches and contexts, in which interventions were reported to be effective or effective. A secondary aim is to summarise participants, family and staff experiences of such interventions.

**Methods and analysis** Ten electronic databases and reference lists from previous similar reviews will be searched. Eligible studies will be process evaluations of RCTs that measure sedentary behaviour as a primary or secondary outcome in adults. As this review will contribute to a programme to develop a community-based intervention to reduce sedentary behaviour in stroke survivors, interventions delivered in schools, colleges, universities or workplaces will be excluded. Two reviewers will perform study selection, data extraction and quality assessment. Disagreements between reviewers will be resolved by a third reviewer. Process evaluation data to be extracted include the aims and methods used in the process evaluation; implementation data; mechanisms of impact; contextual factors; participant, family and staff experiences of the interventions. A narrative approach will be used to synthesise and report qualitative and quantitative data. Reporting of the review will be informed by Preferred Reporting Items for Systematic Review and Meta-Analysis guidance.

**Ethics and dissemination** Ethical approval is not required as it is a protocol for a systematic review. Findings will be disseminated through peer-reviewed publications and conference presentations.

**PROSPERO registration number** CRD42018087403.

## BACKGROUND

Outcome evaluations, such as randomised controlled trials (RCTs), are important to

### Strengths and limitations of this study

► This systematic review protocol follows the Preferred Reporting Items for Systematic Review and Meta-Analysis Protocols guidelines.
► This systematic review addresses a gap in the current evidence-base by providing a comprehensive assessment of the implementation, mechanisms of impact and contextual factors which may influence the effectiveness of randomised controlled trials investigating sedentary behaviour in adults.
► This review is limited to evidence from randomised trials.
► Non-English electronic databases will not be searched. This limitation may cause language bias.
► There is the potential for a low and inconsistent quality in the reporting of process evaluations.

understanding intervention effectiveness, however, in isolation, they may fail to account for how interventions function, why they are successful or not, and for whom.[1] Process evaluations can help to provide this necessary insight.[2] Undertaken alongside outcome evaluations, they include quantitative, qualitative or mixed-methods approaches '*which aim to understand the functioning of an intervention, by examining implementation, mechanisms of impact and contextual factors*'[3] (p8).

Process evaluations may also explore the theoretical and logic models informing or underpinning interventions. A theoretical model may be used by researchers in the development of complex interventions to identify key concepts of interest which may be influential in bringing about a desired outcome or change. Logic models are one method of making theoretical assumptions clear, as they graphically illustrate the link between expected outcomes and

intervention activities/processes designed to bring about these outcomes.[3 4]

### Rationale

Sedentary behaviour is defined as any waking behaviour characterised by low energy expenditure ≤1.5 metabolic equivalents while in a sitting, lying or reclining posture.[5] It has emerged as an important public health issue in the last two decades and has become the focus of considerable clinical, policy and practice research as evidence supporting the detrimental effects of sedentary behaviour on health and well-being has increased.[6–8] The negative impact of sedentary behaviour has been highlighted for a number of parameters related to health,[8 9] including reduced physical function,[10 11] increased symptoms of depression,[12] anxiety[13] and cardiovascular risk.[14 15]

The effectiveness of interventions to reduce sedentary behaviour has been synthesised in systematic reviews and meta-analyses.[16–18] However, such work often fails to provide a detailed understanding of the functioning of the interventions.[19] This systematic review of process evaluations aims to fill this gap in the literature.

This systematic review will contribute to a National Institute for Health Research programme grant for the development and evaluation of strategies to reduce sedentary behaviour in patients after stroke. Currently, there are limited studies looking at reducing sedentary behaviour in stroke survivors, therefore we expanded the search strategy to include all adults to inform a community-based intervention. Although stroke occurs in children and working age adults, the majority of strokes occur in adults aged 65 years and over.[20] Interventions that take place in schools, colleges, universities and workplaces will be excluded from the review as they are less applicable to our population of interest.

### Aims and objectives
#### Review aim

To identify and review previously conducted process evaluations of interventions which include a sedentary behaviour outcome measure in adults, in order to develop an understanding of intervention content, mechanisms of impact, implementation and delivery approaches and contexts, in which interventions were reported to be effective or ineffective. A secondary aim is to explore participant, family and intervention staff experiences in such interventions.

#### Objectives

1. To examine the trial data (eg, design of interventions, sample sizes, duration and content of interventions, and primary and secondary outcome data (from the process evaluation paper or associated papers)).
2. Establish whether logic models or theoretical models were used to explain how interventions were intended to work.
3. Establish whether interventions were delivered as intended.
4. Explore intended or unintended mechanisms of action reported to influence the effectiveness of interventions.
5. Understand barriers and facilitators to delivery of, and participation in, interventions and any recommendations made to address such barriers and facilitators.
6. To examine qualitative data concerning the understanding and experiences of interventions from the perspectives of participants, family/carers and intervention staff.

Qualitative data related to exploring perceptions, views and lived experiences of sedentary behaviour, but *not* related to receipt or delivery of an intervention will be transferred to a concurrent qualitative systematic review (Prospective Register of Systematic Reviews (PROSPERO) registration number: CRD42017083436).

## METHODS AND ANALYSIS

This protocol has been developed following the Preferred Reporting Items for Systematic Review and Meta-Analysis Protocols (PRISMA-P) guidelines,[21] as shown in the PRISMA-P checklist (see online supplementary file 1). The systematic review is prospectively registered with PROSPERO. Reporting of the systematic review will be informed by Preferred Reporting Items for Systematic Review and Meta-Analysis guidance.[22] Important amendments made to the protocol will be documented and published alongside the results of the systematic review.

### Methodological considerations associated with this review
#### Inclusion and exclusion criteria
##### Types of studies

Studies that are explicitly identified as a process evaluation, or studies that aim to understand the functioning of an intervention by examining implementation, mechanisms of impact, and contextual factors[3] (p8) (eg, implementation processes, patient and staff barriers and facilitators, participants' experiences of delivery or receipt of the intervention).

We will include process evaluations of RCTs that measure sedentary behaviour as an outcome in adults. Process evaluations of feasibility RCTs will be included provided there is random allocation. In process evaluations of cross-over trials, we will only include data from the first phase of the trial. Cohort and uncontrolled before-and-after studies will be excluded.

##### Types of participants

All studies involving adults regardless of whether they were conducted in a clinical or nonclinical population. We will include studies with participants aged 16 years or over. We will exclude studies with participants aged <16 years of age.

##### Interventions

Any study which measures sedentary behaviour as an outcome even if reducing sedentary behaviour is not

the primary outcome (eg, moderate-to-vigorous physical activity is the primary outcome).

Interventions that are delivered primarily in schools, colleges, universities or the workplace will be excluded. Studies that do not report any measures of sedentary behaviour as an outcome measure will be excluded. Studies where the main aim is to investigate the acute (immediate) effects of breaking up sitting time as part of a supervised (usually laboratory based) intervention will also be excluded.

### Comparators

In the source trial, the intervention group may be compared with: no active treatment, usual care, attention controls waitlist controls or alternative treatments. Where process evaluations include data from control groups, these data will also be extracted.

### Information sources

#### Electronic searches

In collaboration with information specialist colleagues, informed by guidance from Booth[23] comprehensive search strategies were used using controlled vocabulary and free text terms.

We searched the following electronic databases: CINAHL (EBSCOHost); SPORTDiscus (EBSCOHost); Cochrane Database of Systematic Reviews (Wiley); Cochrane Central Register of Controlled Trials (Wiley): AMED (OVID); EMBASE (OVID); PsycINFO (OVID); Ovid MEDLINE(R); OVID MEDLINE(R) and Epub Ahead of Print, In-Process & Other Non-Indexed Citations; Web of Science: Sciences Citation Index Expanded (Clarivate); Web of Science: Social Sciences Citation Index Expanded (Clarivate); Web of Science: Conference Proceedings Citation Index—Science (Clarivate); Web of Science: Conference Proceedings Citation Index—Social Sciences and Humanities (Clarivate); ProQuest Dissertations & Theses A&I.

#### Searching other relevant sources

In addition to the electronic database searches, we will identify process evaluations through existing systematic reviews of studies of sedentary behaviour interventions. This will include a number of steps:

1. Examining the studies reported within the existing systematic reviews to determine whether they meet the inclusion criteria (eg, randomised trials, adult population, include an outcome measure of sedentary behaviour).
2. We will read the publication of any studies that meet the criteria in step one to identify any process evaluation work. If a process evaluation is referred to, but no data is reported in these publications, we will:
   i. Match the RCTs to any process evaluations identified through the electronic searches (above).
   ii. If they cannot be matched, we will identify linked published process evaluations by performing citation searching (Google Scholar, PubMed and Web

of Science) and also contact authors of the trial publications to request information on any published or unpublished process evaluations.

The above process will also be reversed to match included process evaluation papers with RCTs. This will allow us to extract findings on the intervention outcomes.

A final search syntax for each electronic database is included in online supplementary file 2.

### Study records

#### Data management

We will download references identified in searches (electronic database and additional searches) into Endnote X7 reference management software. Once duplicates are removed, the remaining references will be exported into Covidence (www.covidence.org); an online systematic review tool recommended by the Cochrane Collaboration.

#### Selection process

The screening process will be undertaken using Covidence. Two review authors will independently assess the titles and abstracts of records and exclude papers that do not meet eligibility criteria. We will obtain the full text of the remaining papers, and two authors will assess the papers against the inclusion criteria for the review to determine their eligibility for inclusion. Non-English language papers will be translated into English. The review authors will resolve disagreements through a consensus-based decision, or if necessary, discussion with a third reviewer.

#### Data extraction process

Two review authors will independently extract and record data from included studies using a standardised data extraction form. The data extraction form will be guided by the Medical Research Council guidance for process evaluations,[1] and previous research which has identified key components for conducting and reporting process evaluations.[24 25] Reviewers will pilot the data extraction form with a sample of included papers and amendments will be made as necessary. After piloting, data extraction will be completed using Covidence. Study authors will be contacted if additional information is required. Following data extraction, two reviewers will aim to resolve any discrepancies by a consensus-based decision, or if necessary, discussion with a third reviewer.

We will extract data about the RCT and the process evaluation. Data to be extracted includes:

1. The trial design and trial information:
   a. The number of participants randomised to each group, and demographic information.
   b. The duration and content of what is provided to the intervention group and the comparator group.
   c. Primary and secondary outcome results including adverse events measured at postintervention and follow-up.
2. The aims and objectives of the process evaluation and whether the process evaluation was prespecified or post hoc.

3. The methods used to conduct the process evaluation.
4. The number of sites sampled for the process evaluation, and sample characteristics (eg, recruitment and maintenance of participants or participating sites, reach of the intervention into the target population, age and gender).
5. Implementation data (eg, what is intended to be delivered? How is delivery achieved? What is delivered? How is adherence measured?)
6. Mechanisms of impact (drawing on the logic model or intervention theory used, identified mediators of change, and responses to and interactions with the intervention).
7. Contextual factors that influence implementation, intervention mechanisms and outcomes.
8. Participants, family/carers and intervention staff views and experiences of the interventions, including barriers and facilitators. Experiences of control group participants relating to their involvement in the trial.
9. Any conflicts of interest declared by the authors.

## Outcomes and prioritisation

To meet our research aims and objectives, the outcomes of interest for this study include the following:
1. The outcome results from the intervention.
2. Findings from the process data relating to implementation (intended delivery and fidelity to the intervention plan).
3. Adherence to the intervention and how this is measured.
4. Intended and unintended mechanisms of impact was measured.
5. Barriers and facilitators to delivery or participation in the intervention.
6. Adaptations made to improve delivery of the intervention.
7. Participants experiences of the intervention (delivery and receipt).

Findings will clarify key factors that affect intervention delivery and participation. This will provide contextual information useful for explaining why interventions were effective or ineffective, and how interventions could be refined.

## Quality assessment

Currently, there is no quality assessment tool designed for judging the quality of process evaluations. Process evaluations can incorporate a combination of both qualitative and quantitative data. Therefore, methodological quality will be evaluated using the Mixed Methods Appraisal Tool, which is designed to concurrently assess qualitative, quantitative and mixed-methods studies.[26] Assessment of reporting quality will be guided by Grant *et al*'s framework for reporting process evaluations of cluster RCTs.[25] Two reviewers will independently assess each study and discrepancies will be resolved by a third reviewer. We will not exclude studies based on findings from the quality assessment.

## Data synthesis
### Narrative approach to synthesising data

In this review, we will undertake a narrative approach to synthesising data. The synthesis will provide detailed written commentary on the data extracted in accordance with the factors outlined in the Data collection process section. This will advance our understandings of the intervention context, its delivery, and the mechanisms reported to be effective or ineffective.

## Patient and public involvement

As this research will be based on previously published data, there will be no patient and public involvement in the design, interpretation or dissemination of the findings.

**Author affiliations**
[1]Academic Unit of Elderly Care and Rehabilitation, Bradford Royal Infirmary, Bradford, UK
[2]School of Health Sciences, University of Newcastle, Newcastle, New South Wales, Australia
[3]Clinical Trials Research Unit, University of Leeds, Leeds, UK
[4]ISPEHS, University of Edinburgh, Edinburgh, UK
[5]Academic Unit of Elderly Care and Rehabilitation, University of Leeds, Leeds, UK
[6]Institute of Psychological Sciences, University of Leeds, Leeds, UK
[7]Quality and Safety Research, Bradford Institute for Health Research, Bradford, UK
[8]Geriatric Medicine, University of Edinburgh, Edinburgh, UK

**Acknowledgements** We acknowledge the help and support of our Information Scientist, Deirdre Andre, University of Leeds. We are grateful for the funding provided by the National Institute for Health Research (NIHR).

**Contributors** This systematic review was conceived and designed by members of the Programme Management Group (PMG) of the RECREATE Programme (AF, CE, AFa, CFF, RL, GM, DC). This protocol was initially drafted by DC, RC, JH and AF. Subsequent drafts were commented on by all members of the PMG and revisions made by the authors. All authors have approved submission.

**Funding** This report is independent research funded by the National Institute for Health Research (Programme Grants for Applied Research, Development and evaluation of strategies to reduce sedentary behaviour in patients after stroke and improve outcomes, RP-PG-0615-20019).

**Disclaimer** The views expressed in this publication are those of the author(s) and not necessarily those of the NIHR or the Department of Health and Social Care.

**Competing interests** None declared.

**Patient consent for publication** Not required.

**Provenance and peer review** Not commissioned; externally peer reviewed.

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
