## [Reviewer comments · BMJ Open]

ARTICLE DETAILS

TITLE (PROVISIONAL)	A protocol for a systematic review of process evaluations of interventions investigating sedentary behaviour in adults
AUTHORS	Corepal, Rekesh; Hall, Jessica; English, Coralie; Farrin, Amanda; Fitzsimons, Claire F.; Forster, Anne; Lawton, Rebecca; Mead, Gillian; Clarke, David

VERSION 1 – REVIEW

REVIEWER	Zeljko Pedisic Victoria University, Australia
REVIEW RETURNED	22-May-2019

GENERAL COMMENTS	It was my pleasure to review the protocol entitled “A protocol for a systematic review of process evaluations of interventions investigating sedentary behaviour in adults”. It is a detailed and well-written protocol for a timely and relevant systematic review. Please find my specific, mostly minor terminological, suggestions enclosed below. Page 2 Line 17: Replace “has” with “may have” (because it does not always have) Line 17: Add “an increased risk” (“is associated with an increased risk of”) Line 23: Replace “include” with “included” Lines 23-24: Add “as a primary or secondary outcome” (“... include a measure of sedentary behaviour as a primary or secondary outcome”), because there may be interventions that include a measure of sedentary behaviour as a covariate. Line 29: Replace “explore” with “summarise” Lines 36-37: Replace “included studies from previous similar reviews” with “reference lists from previous similar reviews” (as the current phrasing could be misinterpreted) Lines 41-43: Reasons for this exclusion should be clearly specified in the abstract. This is essential information for understanding your proposal.
--

Lines 55-56: If primary and secondary outcome results are important for this systematic review, they should be searched also outside of the evaluation paper, if they are not presented in the evaluation paper. This should not be an issue, because you will be able to relatively easily find the other publication by searching by the name of the intervention or browsing through author academic profiles. If this is not done, your summary table would be incomplete (for no sound reason).

Page 3

Line 3: Replace “adopted”

Line 33: Replace the title of the document with “Preferred Reporting Items for Systematic review and Meta-Analysis Protocols (PRISMA-P)”. Currently, it is lacking the word “protocols” and the words “reviews” and “meta-analyses” should be in their singular form.

Line 44: This is not quite true, because you will not be conducting searches using keywords translated in all languages. This actually limits your search to publication that have titles, abstracts or keywords in English. I would suggest to delete “language” from here.

General comment: This is a bit odd article summary. It only presents strengths of the article. Please consider rewriting the whole summary. Also, the subtitle “Strengths and limitations of the study” is definitely not matching the content, because no limitations are presented.

Background:

Page 4

Line : Add “may” (“... they may fail to account...”). RCTs can and very often do provide responses to these questions (e.g. “for whom” is an intervention can be found out by analysing interaction effects and conducting stratified analyses).

Lines 50-58: Recent methodological papers suggested that the evidence is not strong, because in most studies the adjustments for sleep, quiet standing, light physical activity, and moderate-to-vigorous physical activity were not done adequately. Furthermore, a few recent methodological papers and xxi studies using pooled datasets would suggest that this conclusion is not right. To avoid getting criticised on this point, I would suggest you not to mention physical activity here. A possible way to deal with this would be to delete both sentences. It is unclear anyway why you emphasised cardiovascular outcomes. Why not diabetes mellitus or some other outcome?

Page 5

Lines 21-22: It is unclear what you mean by “interventions investigating sedentary behaviour”. Are you not focusing only on interventions aimed to tackle sedentary behaviour?

Line 27: Replace “participants” with “participant”

	Line 27: Replace “staffs” with “intervention staff” (note, there is no suffix “s”) Line 29: Replace “of” with “in” Line 34: Replace “participants numbers” with “sample size” Line 38: Please see my previous comment regarding the results of interventions. Line 55: Add “intervention” (“intervention staff”) Page 6 Lines 44-47: Please see my previous comment about including all interventions trials that include a measure of sedentary behaviour. Line 47: Add “uncontrolled” (“... and uncontrolled before-and-after studies will be excluded”). This is essential because RCTs are before-and-after studies as well; yet they are controlled before-and-after studies. This should also be corrected in your PROSPERO registration, if you used this formulation there as well. Line 50: Replace “including both a clinical and nonclinical population” with “regardless whether they were conducted in a clinical or nonclinical population”. Otherwise, it could be misinterpreted in the way that you will only include studies that were conducted in both populations. Line 55: Add “of age” (“16 years of age”) Lines 57-59: This would mean that you have a very loose definition of “adults”. A child is clearly not an adult. I cannot imagine an intervention with such a mixed sample being included in a review of interventions among adults. Please reconsider whether you want include any samples that include children, regardless of the percentage of children in the sample. Also, please define the age brackets for adolescents (or some other way of defining it). This needs to be clarified. Page 7 Lines 3-7: Even if measures of sedentary behaviour are used just as covariates? Lines 9-13: It is unclear why you will not include interventions that are delivered in schools, colleges, universities and workplaces. By excluding those, you would be excluding a great number of interventions. A clear and sound reasoning is needed for this, before this protocol can be accepted. This is essential. Lines 21-32: This should not be presented as a bullet point list. Please reformat. Line 35: Add “also” (“...will also be...”) Line 46: Replace “developed” with “used”, because you have already developed them.
--	--

	Line 48: Please see my previous comment about language restrictions. Line 55 onwards: This should not be presented as a bullet point list. Please reformat. For each bibliographic database, please also specify which search engine will you use (PubMed, EBSCOHost, Web of Science...), as some databases may be searched using different search engines, and the number of search results may vary. Page 8 Line 12: It should be spelled "SPORTDiscus" Line 17: Web of Science is a collection of bibliographic databases. It is essential to specify which Web of Science databases will you search. If you want your search to be more comprehensive, consider also adding Scopus and Academic Search (Premier, Elite, Ultimate or Complete, depending which subscription you have). The process described for matching RCTs identified in systematic reviews with their process evaluation papers should be also reversed to match process evaluation papers found in primary and secondary search with their RCTs (to allow you to extract findings on intervention outcomes). See my previous comments on this. Line 55: A protocol like this has to include a complete and final search syntax for each database. That is one of the main features of protocol for systematic reviews. Writing "draft" here would allow you to tamper with the search syntax at later stages, which is exactly what you are trying to prevent by publishing this protocol. "Any necessary refinements" you mentioned in the following sentence should be done prior to publishing this protocol. Page 9 Line 26: Replace "obviously irrelevant papers" with "papers that do not meet eligibility criteria" Line 31: Replace "foreign language" with "non-English language", because you would be otherwise falsely assuming that English is the only "native" language. Line 31: Add "into English" ("... translated into English.") Line 52: Add a comma after "Following data extraction" Page 10 Line 52: "Intervention staff" Line 22: "was measured" Line 30: Replace "All participants" with "participant"
--	--

	Page 12 Line 20: Replace “secondary” with “previously published” Delete the whole discussion section. This is not a part of a protocol. If the journal requires this section to be included in a protocol for systematic reviews, you should certainly not just reinstate the aims of your protocol. Please refer to guidelines for authors and previous published protocols.
--	--

REVIEWER	Rasmus Tolstrup Larsen Section of Social Medicine, Department of Public Health, University of Copenhagen, Denmark
REVIEW RETURNED	23-May-2019

GENERAL COMMENTS	First of all, I would like to point out that the review is very relevant. However, I have some areas where some revisions could be beneficial for the content. Objectives p. 5 1. This is a bit misleading to read. The review objectives/questions should not be on individual study level. As an example, the objective ii is now a categorical answer on individual study level more suitable for an overview of the individual studies (data extraction). Please use review questions (meta-data) for the objectives. Methods 1. Page 6 line 45. Please be very specific about the study design. You do not want to include RCTs (or feasibility RCTs), but process evaluations of those. Right now, the statement on line 45 indicates otherwise. 2. Page 6, line 56. So, if one study includes 20% participants in ages between 16 and 18, but defined as adults, a similar study includes similar participants, but defined as children, you will have to exclude the latter? I suggest to make a generic lower and upper limit instead. This will be more transparent. 3. Page 7, line 9. Why do you wish to exclude school/workplace interventions? If you wish to exclude these, please state that in an individual sentence. Please do not combine exclusion criteria on setting and outcome measure in one sentence, as you do now. 4. What about project evaluations of cross-over trials? How will you handle this? 5. P.8, line 31. Again, please be very clear about what types of studies you wish to include. 6. Search strategy (additional file 2): under 400 hits in pubmed is not that many. Maybe the Cochrane RCT filter is not that suitable, because you do not want to include RCTs, but process evaluations OF RCTs?
--

	7. Page 9, data management. Covidence is fully capable of handling duplicates. I suggest to use this only. It will also make your reporting easier. Author contribution I have to question the role of 15 authors on the list of a study protocol for a systematic review. I do not find the description of author contributions adequate. ALL authors must fulfill the Vancouver recommendations and, among other, contribute with SUBSTANTIAL work to the conception and design.
--	--

VERSION 1 – AUTHOR RESPONSE

Reviewer 1 comments	Author response to comments
Page2, Line 17: Replace "has" with "may have" (because it does not always have)	Thank you for your suggestion, we have corrected the text as suggested: “has” has been replaced with “may have”
Page2, Line 17: Add "an increased risk" ("is associated with an increased risk of")	Thank you for your comment. We have now added: “an increased risk of”
Page2, Line 23: Replace "include" with "included"	We have now replaced “include” with “included”
Page2, Lines 23-24: Add "as a primary or secondary outcome" ("include a measure of sedentary behaviour as a primary or secondary outcome"), because there may be interventions that include a measure of sedentary behaviour as a covariate.	Thank you for your suggestion we have added the following to provide clarity: “Eligible studies will be process evaluations of RCTs that measure sedentary behaviour as a primary or secondary outcome in adults” has been added
Page2, Line 29: Replace "explore" with "summarise"	As suggested we have now replaced “explore” with “summarise”
Page2, Lines 36-37: Replace "included studies from previous similar reviews" with "reference lists from previous similar reviews" (as the current phrasing could be misinterpreted)	As suggested we have now replaced “included studies from previous similar reviews” with: “reference lists from previous similar reviews”
Page2, Lines 41-43: Reasons for this exclusion should be clearly specified in the abstract. This is essential information for understanding your proposal.	Thank you for your comment. Following your suggestion, we have clarified why interventions delivered in schools, colleges, universities or workplaces will be excluded:

	“As this review will contribute to a programme to develop a community-based intervention to reduce sedentary behaviour in stroke survivors, Interventions delivered in schools, colleges, universities or workplaces will be excluded”. The rationale for these exclusions is expanded upon on page 7 of the manuscript.
Page2, Lines 55-56: If primary and secondary outcome results are important for this systematic review, they should be searched also outside of the evaluation paper, if they are not presented in the evaluation paper. This should not be an issue, because you will be able to relatively easily find the other publication by searching by the name of the intervention or browsing through author academic profiles. If this is not done, your summary table would be incomplete (for no sound reason).	As suggested we have now replaced “if available in the process evaluation paper” with: “if necessary, this information will be sought from related publications linked to the process evaluation paper.”
Page 3, Line 3: Replace "adopted"	As suggested we have now replaced “adopted” with “used” within the text
Page 3, Line 33: Replace the title of the document with "Preferred Reporting Items for Systematic review and Meta-Analysis Protocols (PRISMA-P)". Currently, it is lacking the word "protocols" and the words "reviews" and "meta-analyses" should be in their singular form.	As suggested we have now replaced "Preferred Reporting Items for Systematic reviews and Meta-Analyses (PRISMA-P)" with: "Preferred Reporting Items for Systematic review and Meta-Analysis Protocols (PRISMA-P)"
Page 3, Line 44: This is not quite true, because you will not be conducting searches using keywords translated in all languages. This actually limits your search to publication that have titles, abstracts or keywords in English. I would suggest to delete "language" from here.	As suggested we have now removed reference to “language restrictions”
Page 3: General comment: This is a bit odd article summary. It only presents strengths of the article. Please consider rewriting the whole summary. Also, the subtitle "Strengths and limitations of the study" is definitely not matching the content, because	We agree with the reviewer and now have added the following limitations:
no limitations are presented.	 • This review is limited to evidence from randomised trials • Non-English electronic databases will not be searched. This limitation may cause language bias. • There is the potential for a low and inconsistent quality in the reporting of process evaluations
Page 4, Line 8: Add "may" (" . they may fail to account."). RCTs can and very often do provide responses to these questions (e.g. "for whom" is an intervention can be found out by analysing interaction effects and conducting stratified analyses).	As suggested we have now added “may”

Page 4, Lines 50-58: Recent methodological papers suggested that the evidence is not strong, because in most studies the adjustments for sleep, quiet standing, light physical activity, and moderate-to-vigorous physical activity were not done adequately. Furthermore, a few recent methodological papers and xxi studies using pooled datasets would suggest that this conclusion is not right. To avoid getting criticised on this point, I would suggest you not to mention physical activity here. A possible way to deal with this would be to delete both sentences. It is unclear anyway why you emphasised cardiovascular outcomes. Why not diabetes mellitus or some other outcome?	Thank you for the helpful comment and suggestion. These two sentences have been removed.
Page 5, Lines 21-22: It is unclear what you mean by "interventions investigating sedentary behaviour". Are you not focusing only on interventions aimed to tackle sedentary behaviour?	Thank you for the comment. The focus of the review is on process evaluations of interventions which include a sedentary behaviour outcome. To make this clear "interventions investigating sedentary behaviour" has been replaced by: "interventions which include a sedentary behaviour outcome measure in adults"
Page 5, Line 27: Replace "participants" with "participant"	As suggested we have now replaced "participants" with "participant"
Page 5, Line 27: Replace "staffs" with "intervention staff" (note, there is no suffix "s")	As suggested we have now replaced "staffs" with "staff"
Page 5, Line 29: Replace "of" with "in"	As suggested we have now replaced "of" with "in"
Page 5, Line 34: Replace "participants numbers" with "sample size"	As suggested we have now replaced "participant numbers" with "sample size"
Page 5, Line 38: Please see my previous comment regarding the results of interventions.	As suggested we have now replaced "(if available in the process evaluation paper)" with: ("from the process evaluation paper or associated papers")
Page 5, Line 55: Add "intervention" ("intervention staff")	"intervention" has been added
Page 6, Lines 44-47: Please see my previous comment about including all interventions trials that include a measure of sedentary behaviour.	The focus of the review is on process evaluations of interventions which include a sedentary behaviour outcome. The following sentence has been added: "we will include process evaluations of RCTs that measure sedentary behaviour as an outcome in adults"

Page 6, Line 47: Add "uncontrolled" ("and uncontrolled before-and-after studies will be excluded"). This is essential because RCTs are before-and-after studies as well; yet they are controlled before-and-after studies. This should also be corrected in your PROSPERO registration, if you used this formulation there as well.	As suggested we have now added "uncontrolled"
Page 6, Line 50: Replace "including both a clinical and nonclinical population" with "regardless whether they were conducted in a clinical or nonclinical population". Otherwise, it could be misinterpreted in the way that you will only include studies that were conducted in both populations.	As suggested we have now replaced "including both a clinical and nonclinical population" with: "regardless whether they were conducted in a clinical or nonclinical population"
Page 6, Line 55: Add "of age" ("16 years of age")	As suggested we have now added "of age"
Page 6, Lines 57-59: This would mean that you have a very loose definition of "adults". A child is clearly not an adult. I cannot imagine an intervention with such a mixed sample being included in a review of interventions among adults. Please reconsider whether you want include any samples that include children, regardless of the percentage of children in the sample. Also, please define the age brackets for adolescents (or some other way of defining it). This needs to be clarified.	We had used the existing criteria (page 7) in a previous review but we accept that it could have been clearer and that the criteria could be open to different interpretations. We have revised the statement to say: Types of Participants: All studies involving adults including both a clinical and non-clinical population regardless of whether they were conducted in a clinical or nonclinical population. We will include studies with participants aged 16 or over. We will exclude studies with participants aged less than 16 years of age. We have removed the reference to adolescents.
Page 7, Lines 3-7: Even if measures of sedentary behaviour are used just as covariates?	No, it would be only studies that measure sedentary behaviour as an outcome. To clarify this we have added: "Any study which measures sedentary behaviour as an outcome even if reducing sedentary behaviour is not the primary outcome (e.g. moderate-to-vigorous physical activity (MVPA) is the primary outcome)".
Page 7, Lines 9-13: It is unclear why you will not include interventions that are delivered in schools, colleges, universities and workplaces. By excluding those, you would be excluding a great number of interventions. A clear and sound reasoning is needed for this, before this protocol can be accepted. This is essential.	Thank you for your comment. This systematic review is part of a National Institute for Health Research programme grant for the development and evaluation of strategies to reduce sedentary behaviour in patients after stroke. There are limited studies looking at reducing sedentary behaviour in stroke survivors therefore we expanded the search strategy to include all adults to help inform an intervention. However, we feel that interventions that take place in schools, colleges, universities

	and workplaces will be less applicable to our population of interest (although stroke occurs in children and working age adults, the majority of strokes occur in adults aged 65 and over). To explain the reason why studies conducted in schools, colleges, universities and workplaces will be excluded we have add the following to the introduction section: “This systematic review is part of a National Institute for Health Research programme grant for the development and evaluation of strategies to reduce sedentary behaviour in patients after stroke. Currently, there are limited studies looking at reducing sedentary behaviour in stroke survivors, therefore we expanded the search strategy to include all adults to inform a communitybased intervention. Although stroke occurs in children and working age adults, the majority of strokes occur in adults aged 65 and over.²⁰ interventions that take place in schools, colleges, universities and workplaces will be excluded from the review as they are less applicable to our population of interest”
Page 7, Lines 21-32: This should not be presented as a bullet point list. Please reformat.	Bullet points have been removed
Page 7, Line 35: Add "also" (" .will also be."	As suggested we have now added “also”
Page 7, Line 46: Replace "developed" with "used", because you have already developed them.	As suggested we have now replaced “developed” with “used”
Page 7, Line 48: Please see my previous comment about language restrictions.	Reference to “language restrictions” have been removed
Page 7, Line 55 onwards: This should not be presented as a bullet point list. Please reformat. For each bibliographic database, please also specify which search engine will	We agree with the reviewer and after consultation with our Information Specialist we have reformatted this section as

you use (PubMed, EBSCOHost, Web of Science.), as some databases may be searched using different search engines, and the number of search results may vary.	suggested with the following: “We searched the following electronic databases: CINAHL (EBSCOHost); SPORTDiscus (EBSCOHost); Cochrane Database of Systematic Reviews (Wiley); Cochrane Central Register of Controlled Trials (Wiley); AMED (OVID); EMBASE (OVID); PsycINFO (OVID); Ovid MEDLINE(R); OVID MEDLINE(R) and Epub Ahead of Print, In-Process & Other Non-Indexed Citations; Web of Science: Sciences Citation Index Expanded (Clarivate); Web of Science: Social Sciences Citation Index Expanded (Clarivate); Web of Science: Conference Proceedings Citation Index- Science (Clarivate); Web of Science: Conference Proceedings Citation Index- Social Sciences and Humanities (Clarivate); ProQuest Dissertations & Theses A&I”
Page 8, Line 12: It should be spelled "SPORTDiscus"	“SportDiscus” has been replaced with “SPORTDiscus”
Page 8, Line 17: Web of Science is a collection of bibliographic databases. It is essential to specify which Web of Science databases will you search.	We agree with the reviewer and after consultation with our Information Specialist have reformatted this section as suggested with the following: “Web of Science: Sciences Citation Index Expanded (Clarivate); Web of Science: Social Sciences Citation Index Expanded (Clarivate); Web of Science: Conference Proceedings Citation Index- Science (Clarivate); Web of Science: Conference Proceedings Citation Index- Social Sciences and Humanities (Clarivate)”
Page 8: If you want your search to be more comprehensive, consider also adding Scopus and Academic Search (Premier, Elite, Ultimate or Complete, depending which subscription you have).	Thank you for your suggestion. Unfortunately, we have now completed our electronic database searches but we will be mindful of this for future review work.
Page 8: The process described for matching RCTs identified in systematic reviews with their process evaluation papers should be also reversed to match process evaluation papers found in primary and secondary search with their RCTs (to allow you to extract findings on intervention outcomes). See my previous comments on this.	Thank your suggestion. We have now added the following sentence: “The above process will also be reversed to match included process evaluation papers with RCTs. This will allow us to extract findings on the intervention outcomes”

Page 8, Line 55: A protocol like this has to include a complete and final search syntax for each database. That is one of the main features of protocol for systematic reviews. Writing "draft" here would allow you to tamper with the search syntax at later stages, which is exactly what you are trying to prevent by publishing this protocol. "Any necessary refinements" you mentioned in the following sentence should be done prior to publishing this protocol.	Thank you for the helpful comment. "draft" has been removed. Following your suggestion we have now provided the following clarification: "A final search syntax for each electronic database is included in Additional file 2".
Page 9, Line 26: Replace "obviously irrelevant papers" with "papers that do not meet eligibility criteria"	As suggested we have now replaced "obviously irrelevant papers" with "papers that do not meet eligibility criteria"
Page 9 Line 31: Replace "foreign language" with "non-English language", because you would be otherwise falsely assuming that English is the only "native" language.	As suggested we have now replaced "foreign language" with "non-English language"
Page 9, Line 31: Add "into English" (". translated into English.")	As suggested we have now added "into English"
Page 9, Line 52: Add a comma after "Following data extraction"	Comma has been added
Page 10, Line 52: "Intervention staff"	As suggested we have now replaced "staff" with "intervention staff"
Page 10, Line 22: "was measured"	As suggested we have now added "was measured"
Page 11, Line 30: Replace "All participants" with "participant"	As suggested we have now replaced "All participants" with "participants"
Page 12, Line 20: Replace "secondary" with "previously published"	As suggested we have now replaced "secondary" with "previously published"
Delete the whole discussion section. This is not a part of a protocol. If the journal requires this section to be included in a protocol for systematic reviews, you should certainly not just reinstate the aims of your protocol. Please refer to guidelines for authors and previous published protocols.	Following the helpful suggestion from the reviewer, the discussion section has been removed
Reviewer 2 comments	Author response to comments
Page 5 This is a bit misleading to read. The review objectives/questions should not be on individual study level. As an example, the objective ii is now a categorical answer on individual study level more suitable for an overview of the individual studies (data extraction). Please use review questions (meta-data) for the objectives.	We accept this comment and have revised the wording of each of these objectives so as to make more clear we are not reviewing studies at an individual level (page 7).

Page 6 line 45. Please be very specific about the study design. You do not want to include RCTs (or feasibility RCTs), but process evaluations of those. Right now, the statement on line 45 indicates otherwise.	We agree with the reviewer and have rephrased this section to: “We will include process evaluations of RCTs which include an outcome measure of sedentary behaviour in adults. Process evaluations of feasibility RCTs will be included provided there is random allocation”.
Page 6, line 56. So, if one study includes 20% participants in ages between 16 and 18, but defined as adults, a similar study includes similar participants, but defined as children, you will have to exclude the latter? I suggest to make a generic lower and upper limit instead. This will be more transparent.	We had used the existing criteria (page 7) in a previous review but we accept that it could have been clearer and that the criteria could be open to different interpretations. We have revised the statement to say: Types of Participants: All studies involving adults including both a clinical and non-clinical population regardless of whether they were conducted in a clinical or nonclinical population. We will
	include studies with participants aged 16 or over. We will exclude studies with participants aged less than 16 years of age. We have removed the reference to adolescents.

Page 7, line 9. Why do you wish to exclude school/workplace interventions? If you wish to exclude these, please state that in an individual sentence. Please do not combine exclusion criteria on setting and outcome measure in one sentence, as you do now.	Thank you for your comment. This systematic review is part of a National Institute for Health Research programme grant for the development and evaluation of strategies to reduce sedentary behaviour in patients after stroke. There are limited studies looking at reducing sedentary behaviour in stroke survivors therefore we expanded the search strategy to include all adults to help inform an intervention. However, we feel that interventions that take place in schools, colleges, universities and workplaces will be less applicable to our population of interest. Although stroke occurs in children and working age adults, the majority of strokes occur in adults aged 65 and over. To explain the reason why schools, colleges, universities and workplaces will be excluded we have add the following to the introduction section: “This systematic review is part of a National Institute for Health Research programme grant for the development and evaluation of strategies to reduce sedentary behaviour in patients after stroke. Currently, there are limited studies looking at reducing sedentary behaviour in stroke survivors, therefore we expanded the search strategy to include all adults to inform a communitybased intervention. Although stroke occurs in children and working age adults, the majority of strokes occur in adults aged 65 and over,²⁰ interventions that take place in schools, colleges,
	universities and workplaces will be excluded from the review as they are less applicable to our population of interest” Thank you for your suggestion. Setting and outcome exclusion criteria are now in two separate sentences.

What about project evaluations of cross-over trials? How will you handle this?	Thank you for the comment. We agree that this should be clarified. In cross-over trials, only data from the first phase of the trial will be used to guard against carry over effects. The following sentence has been added to the text for clarification: “in process evaluations of cross-over trials, we will only include data from the first phase of the trial”
P.8, line 31. Again, please be very clear about what types of studies you wish to include.	Thank you for your suggestion. This section has not been amended and we hope provides more clarity. It now reads: “In addition to the electronic database searches, we will identify process evaluations through existing systematic reviews of studies of sedentary behaviour interventions. This will include two steps:  1. Examining the studies reported within the existing systematic reviews to determine whether they meet the inclusion criteria (e.g. randomised trials, adult population, include an outcome measure of sedentary behaviour). 2. We will read the publication of any studies that meet the criteria in step one to identify any process evaluation work”.
Search strategy (additional file 2): under 400 hits in pubmed is not that many. Maybe	Thank you for your comment. We agree that with this filter we
the Cochrane RCT filter is not that suitable, because you do not want to include RCTs, but process evaluations OF RCTs?	will retrieve only RCTs on sedentary behaviour that are process evaluations. However, we feel that the steps described in the text (Searching other relevant sources page 8) will add to the rigour of our searches and help us find studies that may not have been retrieved in the database searches.
Page 9, data management. Covidence is fully capable of handling duplicates. I suggest to use this only. It will also make your reporting easier.	Thank you for your suggestion, we will do this in the future. However, for this review, the electronic database searches have been conducted, and deduped by the information specialist before exporting into Covidence.

Author contribution I have to question the role of 15 authors on the list of a study protocol for a systematic review. I do not find the description of author contributions adequate. ALL authors must fulfil the Vancouver recommendations and, among other, contribute with SUBSTANTIAL work to the conception and design.	Thank you for this comment. We have reviewed the author contributions and have amended the author listing to include those who we consider fulfil the Vancouver requirements and contributed with substantial work to the conception and design of the systematic review protocol. The revised authorship is as follows: Rekesh Corepal (corresponding author); Coralie English Amanda Farrin; Claire Fitzsimons; Anne Forster; Rebecca Lawton; Gillian Mead; David Clarke, on behalf of the RECREATE Programme management Group.
---	--

VERSION 2 – REVIEW

REVIEWER	Zeljko Pedisic Victoria University, Melbourne, Australia
REVIEW RETURNED	25-Jul-2019

GENERAL COMMENTS	Excellent work in addressing reviewer suggestions. I have nothing to add.
---

REVIEWER	Rasmus Tolstrup Larsen Section of Social Medicine, Department of Public Health, University of Copenhagen, Denmark
REVIEW RETURNED	06-Aug-2019

GENERAL COMMENTS	After reading the response letter and reading the revised paper, I recommend the editor to accept the paper. I am satisfied with the revisions to my previous comments.
---